# Functional divergence of paralogous transcription factors supported the evolution of biomineralization in echinoderms

**Jian Ming Khor, Charles A Ettensohn\***

Department of Biological Sciences, Carnegie Mellon University, Pittsburgh, United States

**Abstract** Alx1 is a pivotal transcription factor in a gene regulatory network that controls skeletogenesis throughout the echinoderm phylum. We performed a structure-function analysis of sea urchin Alx1 using a rescue assay and identified a novel, conserved motif (Domain 2) essential for skeletogenic function. The paralogue of Alx1, Alx4, was not functionally interchangeable with Alx1, but insertion of Domain 2 conferred robust skeletogenic function on Alx4. We used cross-species expression experiments to show that Alx1 proteins from distantly related echinoderms are not interchangeable, although the sequence and function of Domain 2 are highly conserved. We also found that Domain 2 is subject to alternative splicing and provide evidence that this domain was originally gained through exonization. Our findings show that a gene duplication event permitted the functional specialization of a transcription factor through changes in exon-intron organization and thereby supported the evolution of a major morphological novelty.

DOI: https://doi.org/10.7554/eLife.32728.001

**\*For correspondence:**
ettensohn@cmu.edu

**Competing interests:** The authors declare that no competing interests exist.

## Introduction

The evolution of animal form has occurred through evolutionary modifications to the developmental programs that give rise to anatomy. There is considerable evidence that evolutionary changes in transcriptional networks have played an important role in morphological evolution (*McGregor et al., 2007*). Furthermore, it is widely accepted that mutations in non-coding, cis-regulatory elements (CREs) have played a major role in the evolution of transcriptional networks (*Rubinstein and de Souza, 2013*; *Wittkopp and Kalay, 2011*). The role of protein-level changes in transcription factors, however, has been more controversial (*Cheatle Jarvela and Hinman, 2015*; *Lynch and Wagner, 2008*). Localized changes in CREs are thought to be more easily tolerated than mutations in coding regions of transcription factors, as the latter could alter the ability of transcription factors to bind to hundreds of CREs throughout the genome and consequently reduce organismal fitness. Mutations in coding regions might be buffered, however, in the event of gene duplication, which frees one copy of the gene from selective pressure, thereby allowing it to evolve novel functions (*Conant and Wolfe, 2008*). Whether functional specialization of paralogous transcription factors has played a significant role in the evolution of novel structures remains an open question.

Echinoderms are a valuable experimental model for addressing the evolution of morphological novelty because of their diverse larval and adult body plans. There are five classes of echinoderms: echinoids (sea urchins), holothuroids (sea cucumbers), asteroids (sea stars), ophiuroids (brittle stars) and crinoids (sea lilies). The echinoids consist of two taxa, the euechinoids (thin-spined sea urchins) and the cidaroids (pencil urchins) (*Telford et al., 2014*). Most echinoderms are maximal indirect developers; that is, they reproduce via a feeding larva which undergoes metamorphosis to produce

the adult. The feeding larvae of echinoids and ophiuroids have extensive, calcite-based endoskeletons, while those of holothuroids have only a rudimentary skeleton, and those of asteroids lack skeletal elements entirely. The feeding larvae of hemichordates, the sister group to echinoderms, also lack a skeleton, suggesting that this structure is derived within the echinoderms. All adult echinoderms, however, regardless of the morphology of their larval forms, possess an extensive, calcitic endoskeleton. Comparative studies have revealed many similarities in the gene regulatory programs of skeletogenic cells in the larva and adult (*Czarkwiani et al., 2013*; *Gao and Davidson, 2008*; *Gao et al., 2015*; *Killian et al., 2010*; *Richardson et al., 1989*). Hence, it is widely thought that the larval skeleton arose within echinoderms by co-option of the adult skeletogenic program.

In sea urchins, the embryonic skeleton is formed by a specialized population of cells known as primary mesenchyme cells (PMCs). These cells are the sole descendants of the large micromeres, four cells that form near the vegetal pole of the cleavage-stage embryo. The specification of the large micromere-PMC lineage occurs during early embryogenesis through a cascade of regulatory (i.e. transcription factor-encoding) genes that are organized into a functional unit termed a gene regulatory network (GRN) (*Ettensohn, 2009*; *Oliveri et al., 2008*). The skeletogenic GRN is activated in the large micromere territory by a combination localized, maternal factors and unequal cell divisions. These maternal inputs activate early regulatory genes, which engage downstream layers of regulatory genes and eventually a suite of several hundred effector genes that control PMC behavior and skeletal morphogenesis (*Ettensohn, 2013*; *Rafiq et al., 2014*).

Alx1 is a pivotal transcription factor in the sea urchin skeletogenic GRN (*Ettensohn et al., 2003*). This protein contains a highly conserved homeodomain, an N-terminal charged region, and a C-terminal OAR domain, which is shared by many Paired-class homeodomain proteins. The precise functions of these various domains (except for the homeodomain) are poorly understood. Throughout embryogenesis, the expression of *alx1* is restricted to the large micromere-PMC lineage. Transcription of the *alx1* gene can be detected as early as the 56 cell stage, during the first cell cycle after the large micromeres are born. The level of *alx1* transcripts peaks at two developmental stages, at the pre-hatching blastula stage and later at the mesenchyme blastula stage, with expression persisting throughout later embryogenesis (*Damle and Davidson, 2011*). Perturbation of *alx1* function using antisense morpholinos (MOs) blocks PMC specification while misexpression of *alx1* results in the ectopic activation of the skeletogenic program in non-PMC lineages (*Ettensohn et al., 2003*; *Ettensohn et al., 2007*). Recently, *alx1* was shown to provide positive inputs into more than 50% of all genes differentially expressed by PMCs, confirming its central role as a regulator of PMC identity and skeletogenesis in sea urchins (*Rafiq et al., 2014*).

Recent comparative studies point to a conserved, essential role for *alx1* in skeletogenesis in all echinoderms. In all echinoderm clades that form larval skeletal elements, *alx1* is activated at an early developmental stage specifically in the skeletogenic lineage (*Dylus et al., 2016*; *Erkenbrack and Davidson, 2015*; *Ettensohn et al., 2003*; *McCauley et al., 2012*; *Rubinstein and de Souza, 2013*). In addition, perturbation experiments have demonstrated that *alx1* is essential for larval skeletogenesis in both pencil urchins and sea cucumbers (*Erkenbrack and Davidson, 2015*; *McCauley et al., 2012*). It is noteworthy that in sea star embryos, which lack a skeleton, *alx1* is expressed at levels that are barely detectable (*Koga et al., 2016*; *McCauley et al., 2012*), although the gene is robustly expressed in skeletogenic centers of adults (*Gao and Davidson, 2008*), as it is in adult skeletogenic centers in sea urchins and brittle stars (*Czarkwiani et al., 2013*; *Gao and Davidson, 2008*; *Gao et al., 2015*). Forced expression of sea urchin or sea star Alx1 in sea stars is sufficient to activate several skeletogenic genes that are also regulated by Alx1 in sea urchins (*Koga et al., 2016*).

In addition to *alx1*, echinoderms possess a paralogous *alx4* gene. The two *alx* genes are located adjacent to one another in the echinoderm genome, suggesting that a gene duplication occurred. A recent molecular phylogenetic analysis by *Koga et al., 2016* showed that both paralogues were present in the last common ancestor of all modern echinoderms and that the gene duplication probably took place after the divergence of echinoderms from hemichordates, which possess a single *alx4*-like gene (*Koga et al., 2016*). In euechinoids, *alx4* is expressed by PMCs but also by non-skeletogenic mesoderm cells (probably presumptive coelomic pouch cells) at the tip of the archenteron (*Koga et al., 2016*; *Rafiq et al., 2012*). The function of Alx4 has not been explored experimentally; however, it has been proposed that because the *alx4*-like gene in hemichordates is expressed in the coelomic mesoderm, the primary function of the ancestral gene was to support coelom

development, a function which may be retained by *alx4* in echinoderms (*Koga et al., 2016*). These findings therefore suggest that the skeletogenic function of *alx1* arose secondarily.

In the present study, we performed a detailed structure-function analysis of sea urchin (*Lytechnius variegatus*) Alx1 (LvAlx1). We developed a rescue assay using morpholino-resistant *alx1* mRNA that allowed us to test the function of motifs in the LvAlx1 protein via mutagenesis. We found that large parts of the protein, including the N-terminal region and most of the C-terminal region, were dispensable for LvAlx1 function in PMC specification, PMC patterning, and skeletogenesis. Further dissection of the C-terminal region, however, revealed that a small, novel domain, which we termed Domain 2, harbored motifs essential for LvAlx1 function. We found that LvAlx4 lacked skeletogenic function by our rescue assay; thus, LvAlx1 and LvAlx4 are not functionally redundant. Remarkably, insertion of LvAlx1 Domain 2 into LvAlx4 conferred robust skeletogenic functions on the LvAlx4 chimeric protein. We found that Domain 2 of Alx1 is contained within a short exon regulated by alternative splicing in sea urchins, pencil urchins and sea stars. Lastly, we used cross-species protein expression experiments to show that while Alx1 proteins from closely related sea urchin species are functionally interchangeable, Alx1 proteins from more distantly related echinoderms are not, due to protein-level differences within the C-terminal region but outside the highly conserved Domain 2. Taken together, our findings support the view that exonization of Domain 2 was an early and critically important event in the functional divergence of Alx1 and Alx4 and a major step in the evolution of echinoderm biomineralization. Other protein-level changes in Alx1 have accumulated more recently and limit the functional interchangeability of Alx1 proteins among echinoderms.

## Results

### A morpholino-resistant form of LvAlx1 rescues LvAlx1 morphants

Orthologues of Alx1 are found in all echinoderms and share a number of highly conserved motifs (*Figure 1*). To perform a structure-function analysis of sea urchin (*L. variegatus*) Alx1, we generated a MO-resistant, GFP-tagged, wild-type *Lv-alx1* mRNA (LvAlx1.WT.GFP) by introducing six silent mutations into the MO target site, which was located at the 5' end of the coding region (*Figure 2A*). When LvAlx1 MO was co-injected with 2 µg/µL of the MO-resistant LvAlx1.WT.GFP mRNA, PMC specification and patterning were substantially rescued (mean percentage of 6a9-positive cells relative to uninjected controls was 99%) (*Figure 2C,C'* and *Figure 2—figure supplement 1*). Because previous work has shown that the effects of Alx1 are dose-dependent (*Ettensohn et al., 2007*), we tested a range of concentrations of LvAlx1.WT.GFP mRNA (1–4 µg/µL) and found that higher concentrations (3–4 µg/µL) inhibited PMC specification, consistent with previous findings. Embryos injected with 3.0 mM LvAlx1 MO showed an almost complete block in PMC specification, as reported previously (*Ettensohn et al., 2003*). Therefore, for all subsequent rescue experiments, we used LvAlx1.WT.GFP mRNA and LvAlx1 MO at concentrations of 2.0 µg/µL and 3.0 mM respectively. For mutant mRNAs of different lengths, mRNA concentrations were adjusted to match the molar concentration of 2.0 µg/µL LvAlx1.WT.GFP.

To rule out the possibility that, despite mutations in the MO target site, LvAlx1.WT.GFP mRNA rescued PMC specification simply by sequestering the MO and preventing it from binding to endogenous LvAlx1 transcripts, we generated a modified LvAlx1.WT.GFP MO-resistant construct that contained multiple, interspersed, premature stop codons (LvAlx1.3STOPS.GFP) (*Figure 2A*). We confirmed by fluorescence microscopy that injection of LvAlx1.3STOPS.GFP mRNA did not yield detectable levels of LvAlx1 protein (*Figure 2—figure supplement 2*). Significantly, embryos co-injected with LvAlx1 MO and LvAlx1.3STOPS.GFP mRNA contained no 6a9-positive cells, confirming that the production of LvAlx1 protein was required for the rescue (mean percentage of 6a9-positive cells relative to uninjected controls was 1.5%) (*Figure 2D,D'* and *Figure 2—figure supplement 1*). We also confirmed that injection of 2.0 µg/µL of LvAlx1.WT.GFP mRNA in the absence of MO induced the formation of supernumerary PMCs (*Figure 2F,F'*), an effect reported by *Ettensohn et al. (2007)*.

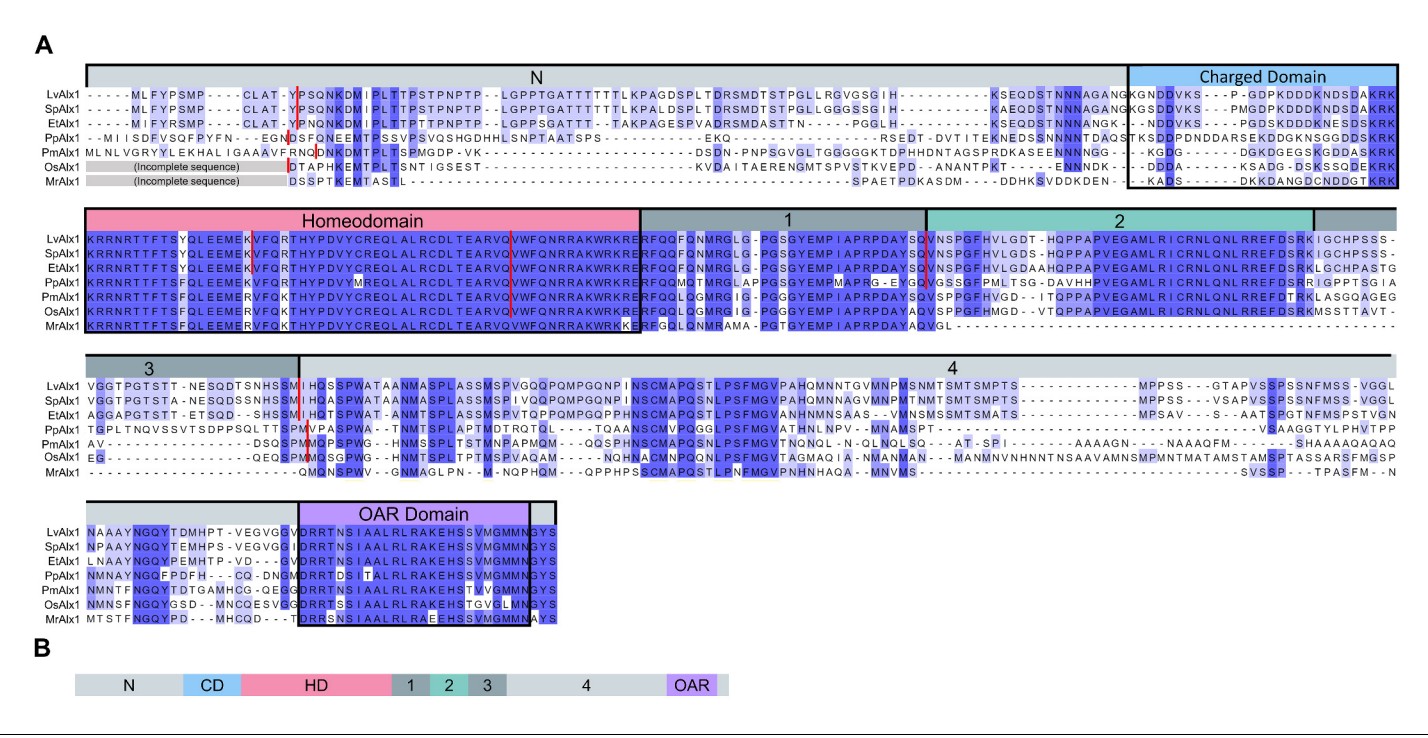

**Figure 1.** Comparison of the predicted amino acid sequences of Alx1 across the echinoderms. (**A**) Clustal Omega alignment of the Alx1 sequences from *Lytechinus variegatus* (LvAlx1), *Strongylocentrotus purpuratus* (SpAlx1), *Eucidaris tribuloides* (EtAlx1), *Parastichopus parvimensis* (PpAlx1), *Patiria miniata* (PmAlx1), *Ophiothrix spiculata* (OsAlx1) and *Metacrinus rotundus* (MrAlx1). The Alx1 proteins share a highly conserved homeodomain (HD) and a C-terminal OAR domain. The region directly upstream of the homeodomain that contains several aspartic acid and lysine residues was designated as the charged domain (CD). The C-terminal region between the homeodomain and OAR domain was divided into four sub-regions, designated Domains 1–4, based on splicing patterns and evolutionary conservation. The degree of sequence similarity is reflected by the intensity of the violet shading. The red lines indicate known splice junctions. As there are no genomic data available for *M. rotundus*, the splice junctions are not known for this species. (**B**) Schematic representation of the domain organization of the echinoderm Alx1 protein.

DOI: https://doi.org/10.7554/eLife.32728.002

## The N-terminal region of LvAlx1 is dispensable for skeletogenic function

We next tested the ability of various mutant forms of LvAlx1 to rescue PMC specification and patterning. In initial experiments, we generated point mutations in a putative MAPK phosphorylation site which was previously identified in the N-terminal region of Alx1 (*Röttinger et al., 2004*). The MAPK signaling pathway has been shown to play an important role in PMC specification and is required for the phosphorylation and activation of Ets1, another important transcription factor expressed by PMCs (*Fernandez-Serra et al., 2004*; *Röttinger et al., 2004*). We altered the putative MAPK phosphorylation site (PSTP) in Alx1 by generating phosphorylation-null (LvAlx1.T28A.GFP) and phosphomimetic (LvAlx1.T28D.GFP) mutants (*Figure 3—figure supplement 1A*). Neither mutation affected the ability of LvAlx1 to rescue morphant embryos (mean percentages of 6a9-positive cells relative to uninjected controls were 97% and 96%, respectively) (*Figure 3—figure supplement 1D–E'* and *Figure 2—figure supplement 1*). These findings suggest that the requirement for MAPK signaling in PMC specification is not mediated through phosphorylation of Alx1.

To more comprehensively examine the role of the N-terminal region, three additional mutants were generated (*Figure 3A*). Deletion of the N-terminal region or charged domain individually revealed that each was dispensable for LvAlx1 function (*Figure 3D–E'*). Surprisingly, even the deletion of all sequences upstream of the homeodomain also did not affect the ability of LvAlx1 to rescue PMC specification and patterning (*Figure 3F,F'*). Thus, based on our rescue assay, the N-terminal region, which constitutes almost one-third of the protein, is dispensable for Alx1 function.

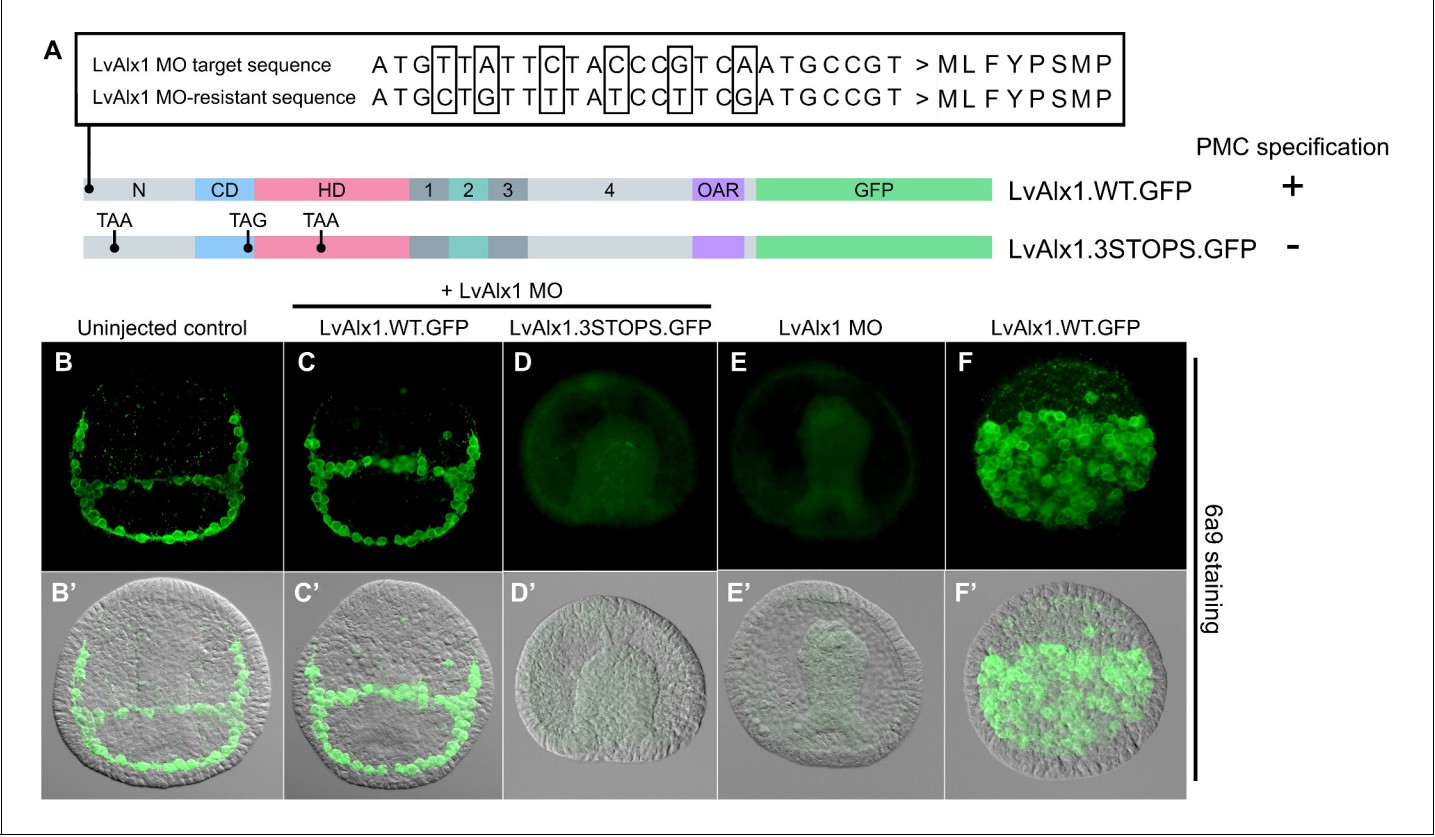

**Figure 2.** Expression of MO-resistant LvAlx1 rescues PMC specification in LvAlx1 morphants. (**A**) Six silent mutations were introduced into the MO target site to generate a MO-resistant *Lv-alx1* mRNA (LvAlx1.WT.GFP). Boxed nucleotides indicate changes introduced. A mutant form of LvAlx1.WT. GFP that contained three in-frame premature stop codons was also generated (LvAlx1.3STOPS.GFP). (**B–F**) Embryos fixed at the mid-late gastrula stage and labelled with a PMC-specific monoclonal antibody (6a9). (**B'–F'**) DIC/fluorescence composite images of the same embryos. (**B, B'**) Uninjected control embryo with 6a9-positive PMCs arranged in a ring-like pattern. (**C, C'**) Embryo co-injected with LvAlx1 MO and LvAlx1.WT.GFP mRNA exhibiting normal PMC specification and patterning. (**D, D'**) Embryo co-injected with LvAlx1 MO and LvAlx1.3STOPS.GFP. This embryo has no 6a9-positive cells, similar to embryos injected with LvAlx1 MO alone (**E, E'**). (**F, F'**) Embryos injected with LvAlx1.WT.GFP mRNA alone showing the induction of supernumerary PMCs.

DOI: https://doi.org/10.7554/eLife.32728.003

The following figure supplements are available for figure 2:

**Figure supplement 1.** Quantitative analysis of MO rescue experiments.
DOI: https://doi.org/10.7554/eLife.32728.004

**Figure supplement 2.** LvAlx1 with premature stop codons, LvAlx1 C region mutants, LvAlx4 and PmAlx1 are expressed at levels similar to that of LvAlx1.WT.GFP.
DOI: https://doi.org/10.7554/eLife.32728.005

This is consistent with the finding that the N-terminal region of Alx1 is not highly conserved among echinoderms.

## Domain 2 of LvAlx1 is a novel functional domain

C-terminal deletion mutations in LvAlx1 were also generated (*Figure 4A*). Surprisingly, removal of the highly conserved OAR domain (LvAlx1.ΔOAR.GFP) did not markedly affect rescue of LvAlx1 morphants (mean percentage of 6a9-positive cells relative to uninjected controls was 86%) (*Figure 4D, D'* and *Figure 2—figure supplement 1*). By contrast, deletion of the C-terminal region between the homeodomain and OAR domain (LvAlx1.ΔC.GFP) almost completely abolished rescue (mean percentage of 6a9-positive cells relative to uninjected controls was 1.3%) (*Figure 4E,E'* and *Figure 2—figure supplement 1*), similar to the total truncation of the C-terminus (LvAlx1.ΔCΔOAR.GFP) (mean percentage of 6a9-positive cells relative to uninjected controls was 2.0%) (*Figure 4F,F'* and

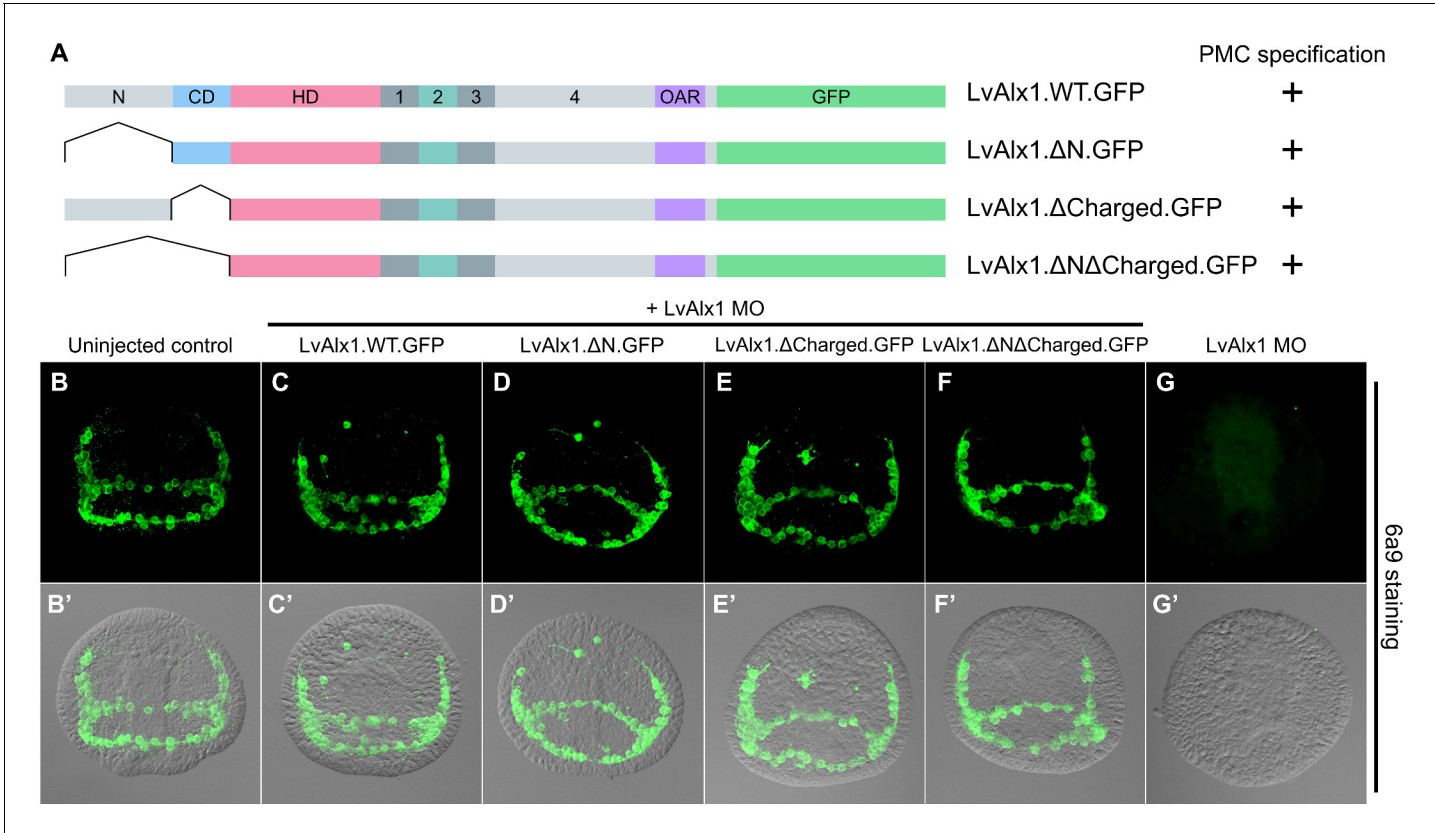

**Figure 3.** The N-terminus is dispensable for LvAlx1 function in PMC specification and patterning. (**A**) Constructs generated with different N-terminal deletions and truncations. (**B–G**) Embryos fixed at the mid-late gastrula stage and labelled with 6a9 antibody. (**B'–G'**) DIC/fluorescence composite images of the same embryos. (**B, B'**) Uninjected control embryo. (**C–F'**) Embryos co-injected with LvAlx1 MO and LvAlx1 mRNA with different N-terminal mutations exhibit normal PMC specification and patterning, similar to the positive control embryo co-injected with LvAlx1 MO and LvAlx1. WT.GFP mRNA (**C, C'**). Total deletion of the region upstream of the homeodomain (LvAlx1.ΔNΔCharged.GFP) also does not affect PMC specification or patterning (**F, F'**). (**G, G'**) Embryo injected with LvAlx1 MO alone (negative control).
DOI: https://doi.org/10.7554/eLife.32728.006

The following figure supplement is available for figure 3:

**Figure supplement 1.** The putative MAPK phosphorylation site T28 is not essential for PMC specification or patterning.
DOI: https://doi.org/10.7554/eLife.32728.007

*Figure 2—figure supplement 1*). These observations showed that the C-terminal region contained motifs essential for Alx1 function and prompted further dissection of this region.

To identify possible functional domains in the C-terminal region, the sequence between the homeodomain and OAR domain was divided into four regions, designated Domains 1–4, based on splicing patterns and evolutionary conservation (*Figure 1A*). In initial studies, we found that the large Domain 4 was dispensable for skeletogenic function (LvAlx1.ΔD4.GFP) (*Figure 5D,D'*), while deletion of a region spanning Domains 1–3 abolished rescue (LvAlx1.ΔD123.GFP) (*Figure 5E,E'*). Subsequently, we subdivided the Domain 1–3 region into separate domains. Deletion of Domain 2 alone almost completely abolished rescue (LvAlx1.ΔD2.GFP) (mean percentage of 6a9-positive cells relative to uninjected controls was 7.5%) (*Figure 5G,G'* and *Figure 2—figure supplement 1*). By contrast, deletion of Domain 1 or Domain 3 individually did not affect LvAlx1 function by this assay (LvAlx1.ΔD1.GFP and LvAlx1.ΔD3.GFP) (mean percentages of 6a9-positive cells relative to uninjected controls were 100% and 101%, respectively) (*Figure 5F,F',H,H'* and *Figure 2—figure supplement 1*). These observations therefore pointed to the small (41-amino acid) Domain 2 as being of special importance in supporting skeletogenesis.

Taken together, our deletion studies revealed that large regions of the Alx1 protein, including the N-terminus, the charged domain, Domains 1, 3 and 4, and the OAR domain, were dispensable

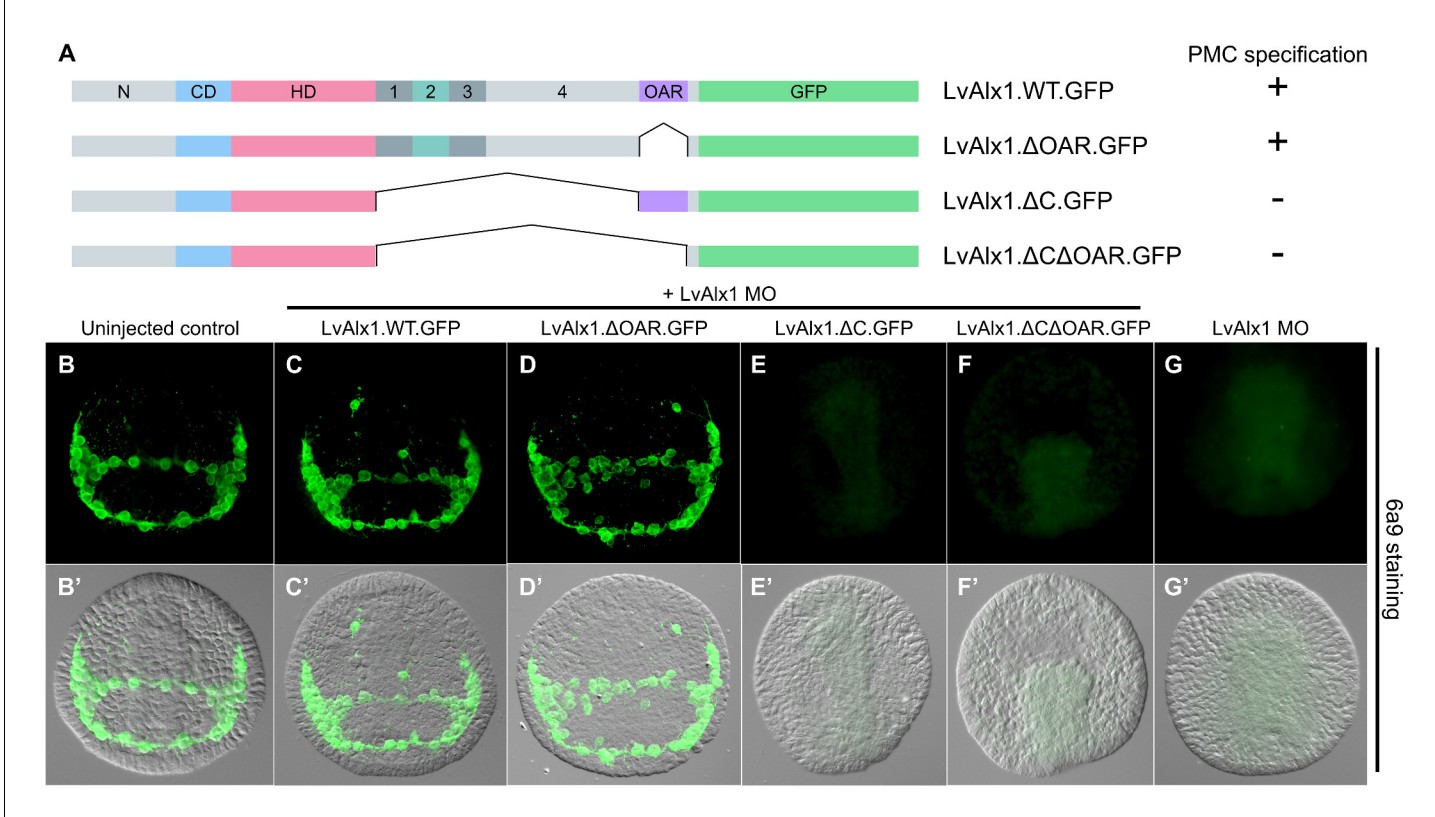

**Figure 4.** The C-terminal region contains motifs essential for Alx1 function. (**A**) Constructs generated with different C-terminal deletions and truncations. (**B–G**) Embryos fixed at the mid-late gastrula stage and labelled with 6a9 antibody. (**B′–G′**) DIC/fluorescence composite images of the same embryos. (**B, B′**) Uninjected control embryo. (**C, C′**) Positive control embryo co-injected with LvAlx1 MO and LvAlx1.WT.GFP. (**D, D′**) Deletion of the highly conserved OAR domain does not affect rescue of PMC specification or patterning. (**E, E′ and F, F′**) Embryos co-injected with LvAlx1 MO and LvAlx1 mRNA lacking C-terminal sequences (LvAlx1.ΔC.GFP and LvAlx1.ΔC.ΔOAR.GFP) contain no 6a9-positive cells, similar to embryos injected with LvAlx1 MO alone (**G, G′**).

DOI: https://doi.org/10.7554/eLife.32728.008

for function in the embryo. To define a minimal construct sufficient for skeletogenic function, we generated three mutants with progressively larger truncations of the C-terminal region (*Figure 6A*). We observed that a minimal construct containing only the homeodomain and Domains 1 and 2 was sufficient to completely rescue PMC specification and skeletogenesis (mean percentage of 6a9-positive cells relative to uninjected controls was 98%) (*Figure 6F,F′* and *Figure 2—figure supplement 1*). This construct encoded a polypeptide only 132 amino acids in length; that is, less than 1/3 the length of wild-type Alx1 (429 amino acids).

A potential concern regarding our rescue assay was that there might be variability in the levels of expression of different mutant Alx1 proteins or in their subcellular localization. We confirmed by fluorescence microscopy that mutant Alx1 proteins were expressed at levels very similar to that of LvAlx1.WT.GFP construct (*Figure 2—figure supplement 2*). We also confirmed that all mutant proteins, like wild-type Alx1, were localized to the cell nucleus. We analyzed the nuclear localization signals (NLS) of LvAlx1 and found that the homeodomain was flanked by NLS motifs that were necessary and sufficient to drive nuclear localization (*Figure 6—figure supplement 1*). Other homeodomain proteins also harbor NLS sequences at these same locations (*Shoubridge et al., 2010*; *Ye et al., 2011*). Both NLS motifs were present in all Alx1 mutant constructs that were tested by the MO rescue assay.

We confirmed that rescue of the morphant phenotype was not limited to early PMC specification and early patterning but was also associated with a full rescue of skeletal morphogenesis. For example, embryos co-injected with LvAlx1 MO and LvAlx1.ΔNΔChargedΔOARΔD34.GFP had normal

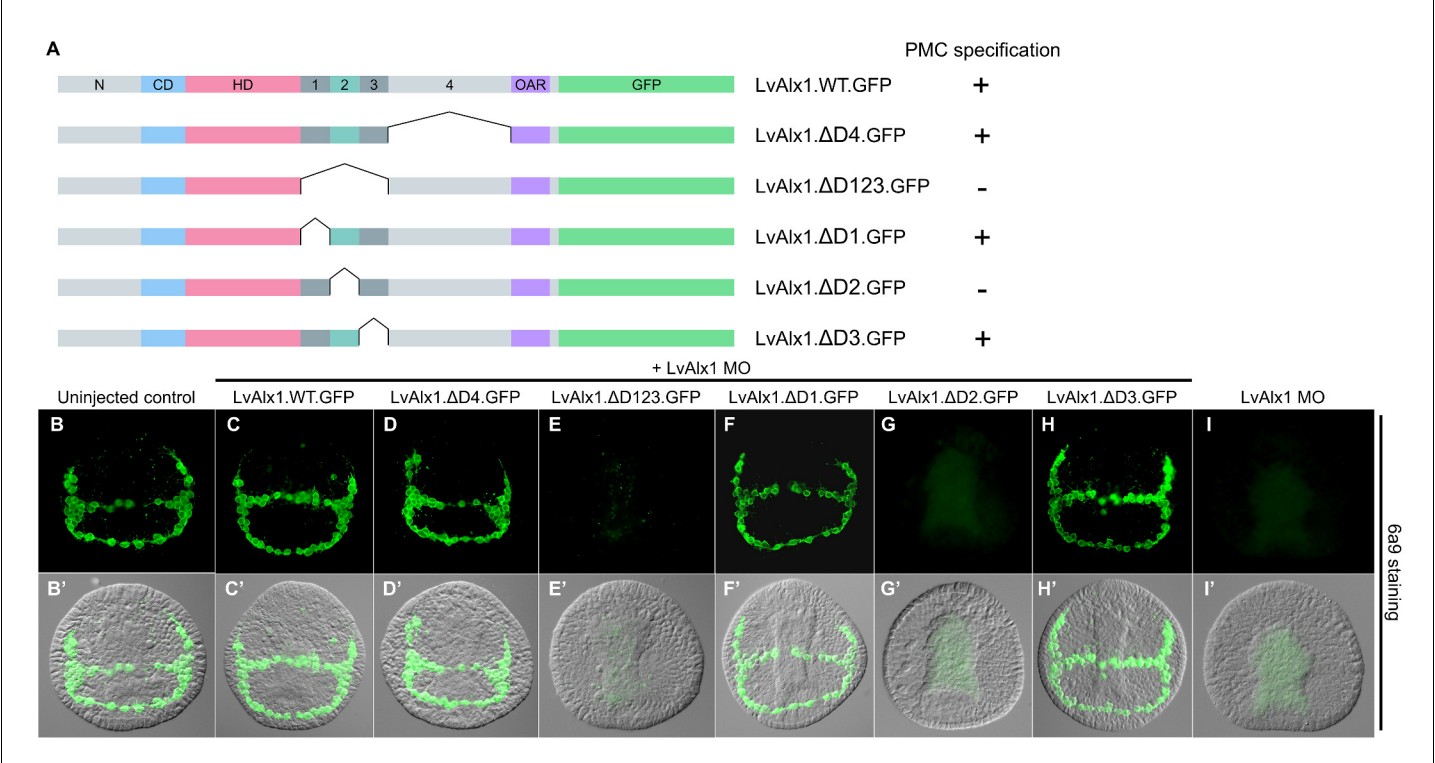

**Figure 5.** Domain 2 is essential for Alx1 function. (**A**) Constructs with deletions within the C-terminal region. (**B–I**) Embryos fixed at the mid-late gastrula stage and labelled with 6a9 antibody. (**B'–I'**) DIC/fluorescence composite images of the same embryos. (**B, B'**) Uninjected control embryo. (**C, C'**) Positive control embryo co-injected with LvAlx1 MO and LvAlx1.WT.GFP. (**D, D'**) Deletion of the relatively large Domain 4 does not affect rescue of PMC specification and patterning. (**E, E'**) Deletion of Domains 1–3 together abolishes rescue, as does deletion of Domain 2 alone (**G, G'**). (**F, F' and H, H'**) Deletion of Domains 1 or 3 individually does not affect rescue. (**I, I'**) Embryo injected with LvAlx1 MO alone (negative control).
DOI: https://doi.org/10.7554/eLife.32728.009

triradiate spicules at the prism stage and elongated skeletal rods at the pluteus stage (**Figure 6—figure supplement 2C,I**), confirming that a minimal construct containing just the homeodomain and Domains 1 and 2 was sufficient for complete rescue of LvAlx1 morphants.

## LvAlx1 and LvAlx4 are not functionally redundant, but experimental inclusion of Domain 2 confers skeletogenic function on LvAlx4

The paralogous LvAlx1 and LvAlx4 proteins share an almost identical homeodomain and a highly conserved OAR domain while the remaining protein sequences are mostly dissimilar (**Figure 7A**). In particular, LvAlx4 lacks Domain 2. We found that LvAlx4.WT.GFP mRNA was unable to replace LvAlx1 in specifying PMCs (mean percentage of 6a9-positive cells relative to uninjected controls was 5.3%) (**Figure 7E,E'** and **Figure 2—figure supplement 1**). We confirmed by fluorescence microscopy that the LvAlx4.WT.GFP protein accumulated in the nucleus and was expressed at similar levels as LvAlx1.WT.GFP (**Figure 2—figure supplement 2**). To determine whether motifs from LvAlx1 can confer early skeletogenic function on LvAlx4, we generated a series of chimeric proteins, with LvAlx4 as the backbone. Replacement of the region between the homeodomain and OAR domain of LvAlx4 with the corresponding region of LvAlx1 (LvAlx4.LvAlx1C.GFP) resulted in a striking rescue of PMC specification and patterning (mean percentage of 6a9-positive cells relative to uninjected controls was 87%) (**Figure 7F,F'** and **Figure 2—figure supplement 1**). The minimum region that was sufficient to confer early skeletogenic function was subsequently narrowed down to Domain 2 (LvAlx4.LvAlx1D2.GFP) (mean percentage of 6a9-positive cells relative to uninjected controls was 96%) (**Figure 7G–H'** and **Figure 2—figure supplement 1**). We also showed that the chimeric LvAlx4 protein containing LvAlx1 Domain 2 fully rescued later skeletal morphogenesis (**Figure 6—figure supplement 2D,K**). These findings further confirmed the importance of Domain 2 in LvAlx1 function. In

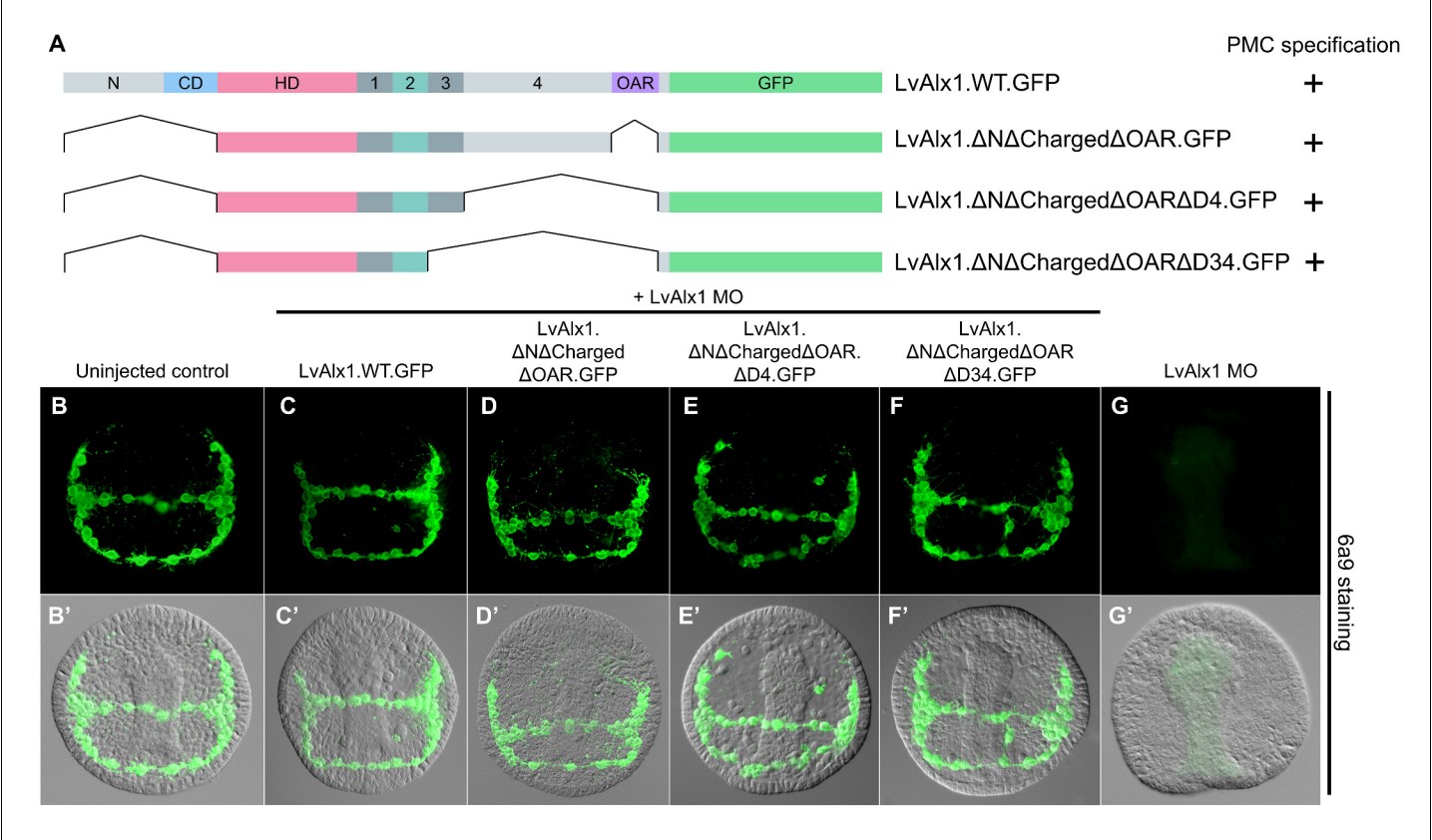

**Figure 6.** Determination of a minimal construct sufficient to rescue LvAlx1 morphants. (**A**) Constructs tested. All mutant constructs contained the homeodomain and Domain 2, both of which were required for PMC specification. (**B–G**) Embryos fixed at the mid-late gastrula stage and labelled with 6a9 antibody. (**B'–G'**) DIC/fluorescence composite images of the same embryos. (**B, B'**) Uninjected control embryo. (**C, C'**) Positive control embryo co-injected with LvAlx1 MO and LvAlx1.WT.GFP (**D–F'**) LvAlx1 mRNA that lacked the N-terminus, charged domain, OAR domain and Domains 3 and 4 (LvAlx1.ΔNΔChargedΔOARΔD34.GFP) was sufficient to rescue LvAlx1 morphants. (**G, G'**) Embryo injected with LvAlx1 MO alone (negative control).
DOI: https://doi.org/10.7554/eLife.32728.010

The following figure supplements are available for figure 6:

**Figure supplement 1.** Basic residues flanking the LvAlx1 homeodomain function as nuclear localization signals.
DOI: https://doi.org/10.7554/eLife.32728.011

**Figure supplement 2.** Rescued embryos show normal skeleton development at late embryonic stages.
DOI: https://doi.org/10.7554/eLife.32728.012

addition, they suggested that the gain of this domain might have been a critical innovation that marked the functional specialization of the ancestral *alx1* gene.

## Alx1 orthologues from closely related sea urchin species, but not from more distantly related echinoderms, are functionally interchangeable

The Alx1 protein from *Strongylocentrotus purpuratus* (SpAlx1), a sea urchin species relatively closely related to *L. variegatus* (divergence time ~100 million years), has an amino acid sequence almost identical to that of LvAlx1, while the Alx1 protein from a more distantly related echinoderm, the sea star *Patiria miniata* (PmAlx1) (divergence time >450 million years) is less well conserved (*Figure 1A*). To investigate whether these evolutionary changes in Alx1 sequence are associated with changes in the functional properties of the protein, LvAlx1 MO and mRNA encoding either *S. purpuratus* Alx1 (SpAlx1) or *P. miniata* Alx1 (PmAlx1) were co-injected into fertilized eggs. SpAlx1 was interchangeable with LvAlx1 but PmAlx1 was not (*Figure 8D–E'*). The PmAlx1.WT.GFP protein showed nuclear accumulation and robust expression, similar to LvAlx1.WT.GFP (*Figure 2—figure supplement 2*). It should also be noted that PmAlx1 possesses its own Domain 2. We found that the protein-level changes in PmAlx1 that limited its function in the sea urchin embryo were located within the

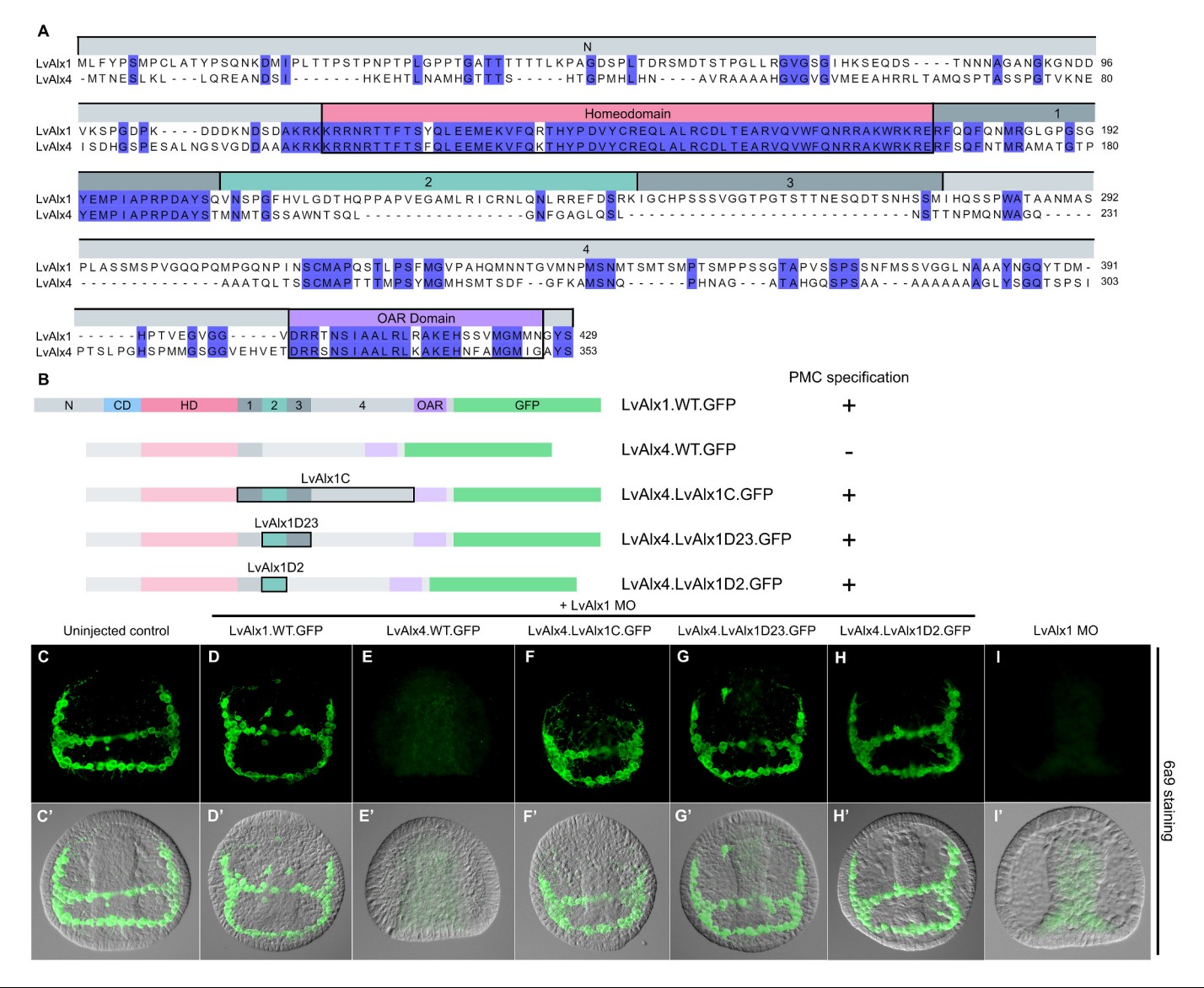

**Figure 7.** LvAlx1 and LvAlx4 are not functionally redundant; however, insertion of LvAlx1 Domain 2 into LvAlx4 is sufficient to confer skeletogenic function. (A) Clustal Omega alignment of the predicted amino acid sequences of LvAlx4 and LvAlx1. (C–I) Embryos fixed at the mid-late gastrula stage and labelled with 6a9 antibody. (C'–I') DIC/fluorescence composite images of the same embryos. (C, C') Uninjected control embryo. (D, D') Positive control embryo co-injected with LvAlx1 MO and LvAlx1.WT.GFP. (E, E') Embryo co-injected with LvAlx1 MO and LvAlx4 mRNA showing no 6a9-positive cells. (F, F', G, G' and H, H') Chimeric forms of LvAlx4 containing the entire C-terminal region of LvAlx1, Domains 2 and 3, or Domain 2 alone, are able to rescue PMC specification and patterning. (I, I') Embryo injected with LvAlx1 MO alone (negative control).
DOI: https://doi.org/10.7554/eLife.32728.013

C-terminal region between the homeodomain and OAR domain, as a chimeric form of PmAlx1 containing this region of the LvAlx1 protein (PmAlx1.LvAlx1C.GFP) rescued PMC specification and patterning in LvAlx1 morphants (*Figure 8,F'*). These embryos also exhibited normal skeletal development at later developmental stages (*Figure 6—figure supplement 2E,L*). To determine whether PmAlx1 Domain 2 was functionally interchangeable with LvAlx1 Domain 2, we created an LvAlx1 construct in which the endogenous Domain 2 was replaced by PmAlx1 Domain 2 (LvAlx1.PmAlx1D2.GFP). This chimeric protein efficiently rescued PMC specification and patterning in LvAlx1 morphants (*Figure 8,G'*; *Figure 2—figure supplement 1*). These findings highlighted the striking functional conservation of Domain 2 across >450 million years of evolution. At the same time, they

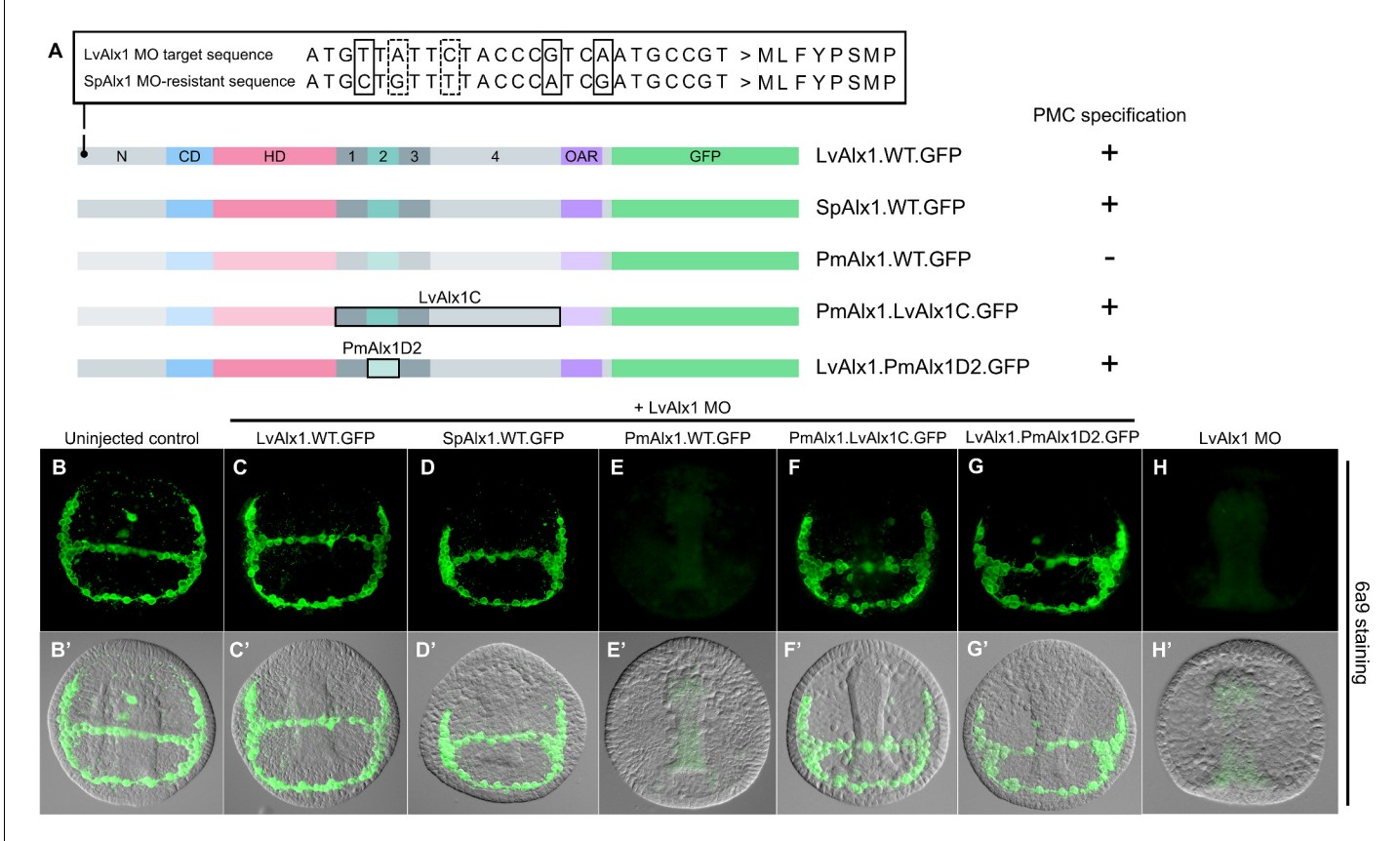

**Figure 8.** Alx1 proteins from closely related echinoderm species, but not from distantly related species, are functionally interchangeable with LvAlx1. (**A**) Three silent mutations (solid boxes) were introduced into the *S. purpuratus alx1* translational start site to generate a form of *Sp-alx1* mRNA (SpAlx1. WT.GFP) that was resistant to the LvAlx1 MO. Dashed boxes indicate natural nucleotide differences between *Lv-alx1* and *Sp-alx1*. The *P. miniata alx1* translational start site is very different from that of *Lv-alx1* and hence no mutations were necessary. (**B–G**) Embryos fixed at the mid-late gastrula stage and labelled with 6a9 antibody. (**B'–G'**) DIC/fluorescence composite images of the same embryos. (**B, B'**) Uninjected control embryo. (**C, C'**) Positive control embryo co-injected with LvAlx1 MO and LvAlx1.WT.GFP. (**D, D'**) Alx1 from a closely related species, the sea urchin *S. purpuratus* (SpAlx1), rescues PMC specification and patterning in LvAlx1 morphants. (**E, E'**) Alx1 from a more distantly related species, the sea star *P. miniata* (PmAlx1), is not interchangeable with LvAlx1. (**F, F'**) A chimeric form of PmAlx1 containing the C-terminal region of LvAlx1 between the homeodomain and OAR domain (PmAlx1.LvAlx1C.GFP) rescues PMC specification and patterning in LvAlx1 morphants. (**G, G'**). A chimeric form of LvAlx1 which had the endogenous Domain 2 replaced with Domain 2 of PmAlx1 (LvAlx1.PmAlx1D2.GFP) rescues PMC specification. (**H, H'**) Embryo injected with LvAlx1 MO alone (negative control).

DOI: https://doi.org/10.7554/eLife.32728.014

showed that the inability of PmAlx1 to support skeletogenesis in sea urchins was attributable to sequence differences within the C-terminal region but outside the conserved Domain 2.

## Domain 2 is subject to alternative splicing and likely arose by exonization

Although the protein sequences of echinoderm Alx1 orthologues are very similar to one another, the intron-exon structures of the cognate *alx1* genes and the splicing patterns of their primary transcripts show considerable variability. We analyzed the intron-exon organization of echinoderm *alx1* and *alx4* genes using publicly available genomic and cDNA sequences (see Materials and methods) (*Figure 9*). This analysis showed that the intron-exon organization of the *alx4* genes in all echinoderm clades has been strictly conserved and corresponds closely to that of the single hemichordate *alx4*-like gene. By contrast, the intron-exon organization of echinoderm *alx1* genes is more variable and in all clades these genes contain greater numbers of exons than their *alx4* paralogues. These findings support the view that the duplication of the ancestral *alx4*-like gene was followed by an

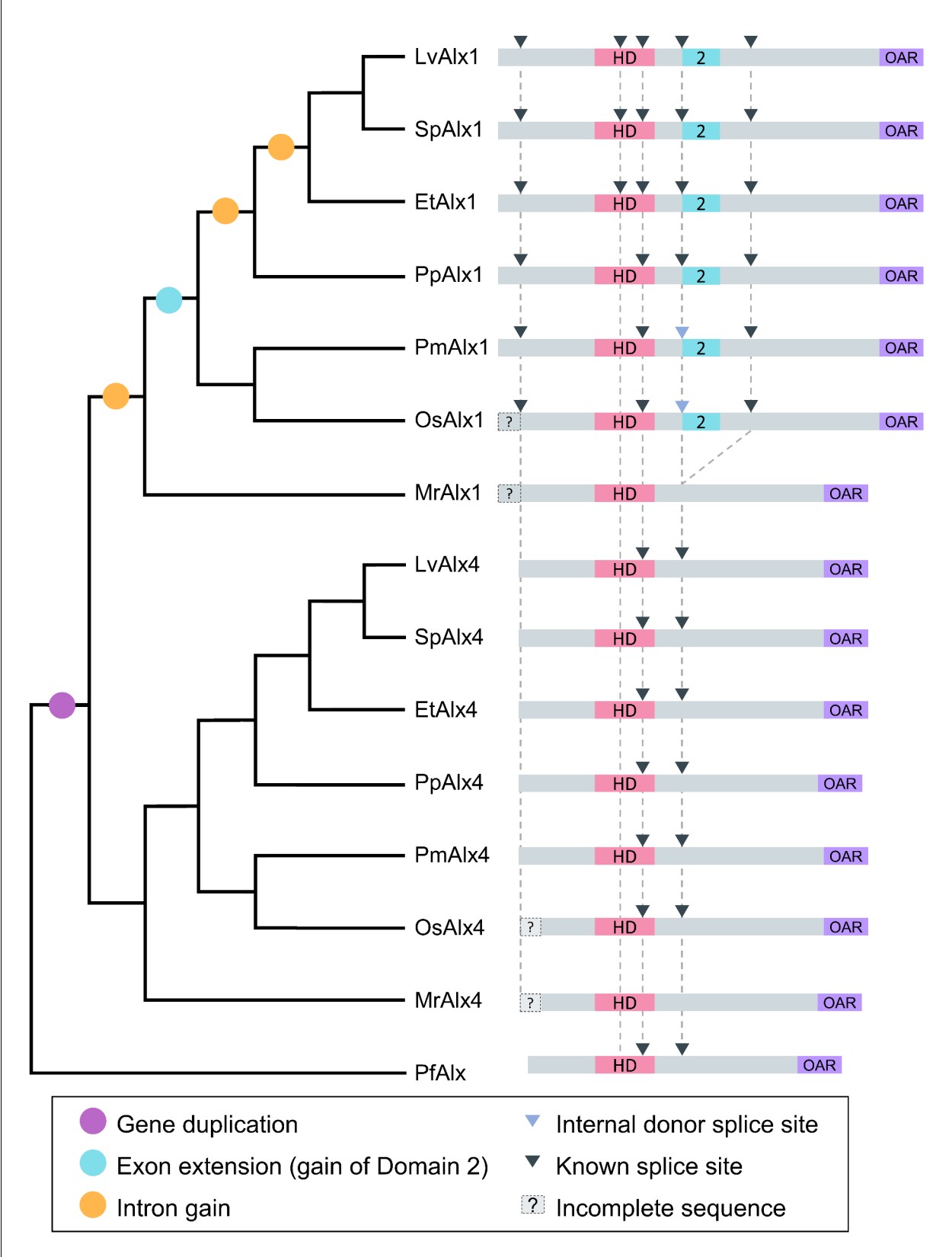

**Figure 9.** Evolution of echinoderm *alx* gene structure. Left side: molecular phylogeny of echinoderm Alx1 and Alx4 proteins (based on ***Koga et al., 2016***), with a hemichordate as an outgroup. Branch lengths are arbitrary. Right side: intron-exon organization of echinoderm *alx1* and *alx4* genes. Lv:

*Figure 9 continued on next page*

*Figure 9 continued*

*Lytechinus variegatus* (euchinoid sea urchin); Sp: *Strongylocentrotus purpuratus* (euchinoid sea urchin); Et: *Eucidaris tribuloides* (cidaroid sea urchin); Pp: *Parastichopus parvimensis* (sea cucumber); Pm: *Patiria miniata* (sea star); Os: *Ophiothrix spiculata* (brittle star); Mr: *Metacrinus rotundus* (sea lily); Pf: *Ptychodera flava* (hemichordate). Splice sites for *Mr-alx1* and *Mr-alx4* are not known. Dotted lines are for alignment purposes only.

DOI: https://doi.org/10.7554/eLife.32728.015

increase in the structural complexity of the paralogue that was the progenitor of the modern *alx1* gene.

Nucleotide and protein BLAST searches did not identify Domain 2-like sequences in genes other than *alx1*, either within or outside of the echinoderms, supporting the view that this domain arose de novo within the *alx1* gene by the recruitment of non-coding sequences. It has been shown that intrinsically disordered regions are enriched in domains obtained through exonization of intronic sequences (*Buljan et al., 2010*). Using InterPro protein sequence analysis and the MobiDB database of intrinsically disordered and mobile proteins, we found that the Domain 3 regions of LvAlx1, SpAlx1, EtAlx1, PpAlx1, PmAlx1 and OsAlx1 are all predicted to be intrinsically disordered, despite their divergent sequences (*Finn et al., 2017*; *Potenza et al., 2015*). These observations are consistent with the hypothesis that the Domain (2 + 3) exon arose through the recruitment of previously non-coding sequences.

Newly born exons are alternatively spliced at a much higher frequency than old exons, and it has been argued that alternative splicing may provide an opportunity for natural selection to act on new protein structures without completely eliminating the parental form (*Sorek, 2007*). We therefore examined the pattern of *alx1* splicing in the vicinity of the Domain (2 + 3) exon. In echinozoan *alx1* genes (*Lv-alx1*, *Sp-alx1*, *Et-alx1* and *Pp-alx1*), Domains 2 and 3 are contained on a single, separate exon. We analyzed the pattern of splicing in this region by RT-PCR and found that at all developmental stages tested, the wild-type [i.e. Domain (2 + 3)-containing] *Lv-alx1* mRNA was the predominant isoform (*Figure 10A and B*). Two minor splice variants were also detected, and these were extracted and sequenced. In LvAlx1 splice variant 1, an alternative splice donor site at the 3' end of Domain 3 was utilized, introducing a frameshift and resulting in a premature stop codon within Domain 4. In LvAlx1 splice variant 2, the exon containing Domains 2 and 3 was skipped. These results closely matched those obtained by RNA-seq, which also showed one major splicing isoform that contained the Domain (2 + 3) exon and low levels of an isoform lacking this exon (*Rafiq et al., 2014*). In addition, in the course of cloning and sequencing *alx1* cDNAs from the pencil urchin, *Eucidaris tribuloides*, we identified a mixture of clones that either contained or lacked the Domain (2 + 3) exon, indicating that this region was also alternatively spliced in this species (data not shown).

In contrast to echinozoan *alx1* genes, in an asterozoan, Domains 2 and 3 of Alx1 are encoded by a continuous open reading frame that also includes a portion of the homeodomain (*Figure 10C and D*). We noted that asterozoan *alx1* genes contain a potential splice site at the 5' end of Domain 2 that corresponds perfectly to the 5' splice junction of the echinozoan *alx1* genes. RT-PCR analysis of RNA extracted from *P. miniata* embryos at several developmental stages revealed five major splice variants in this region. Sequencing results revealed that, in addition to the 'wild-type' mRNA (which we defined as the isoform that was most similar to the abundant isoform in sea urchins), PmAlx1 splice variant 1 contained an in-frame, alternative exon between Domains 3 and 4. In PmAlx1 splice variant 2, an alternative splice site at the 3' end of Domain 3 was used to join the alternative exon, resulting in a frameshift and premature stop codon. In PmAlx1 splice variant 3, the alternative exon took the place of Domains 2 and 3, which were spliced out using an internal splice site at the 5' end of Domain 2. In PmAlx1 splice variant 4, the Domain (2 + 3) exon and alternative exon were skipped. We also noted a change in the splicing pattern across development, such that by the late bipinnaria stage, when adult skeletal elements are being produced, levels of the wild-type PmAlx1 which contained the Domain (2 + 3) exon were elevated. These findings revealed a complex pattern of alternative splicing of the *alx1* transcript in *P. miniata* and showed that the splicing pattern is developmentally regulated, with a shift to a greater abundance of the wild-type mRNA isoform (which encodes Domain 2) at stages when skeletogenesis is underway.

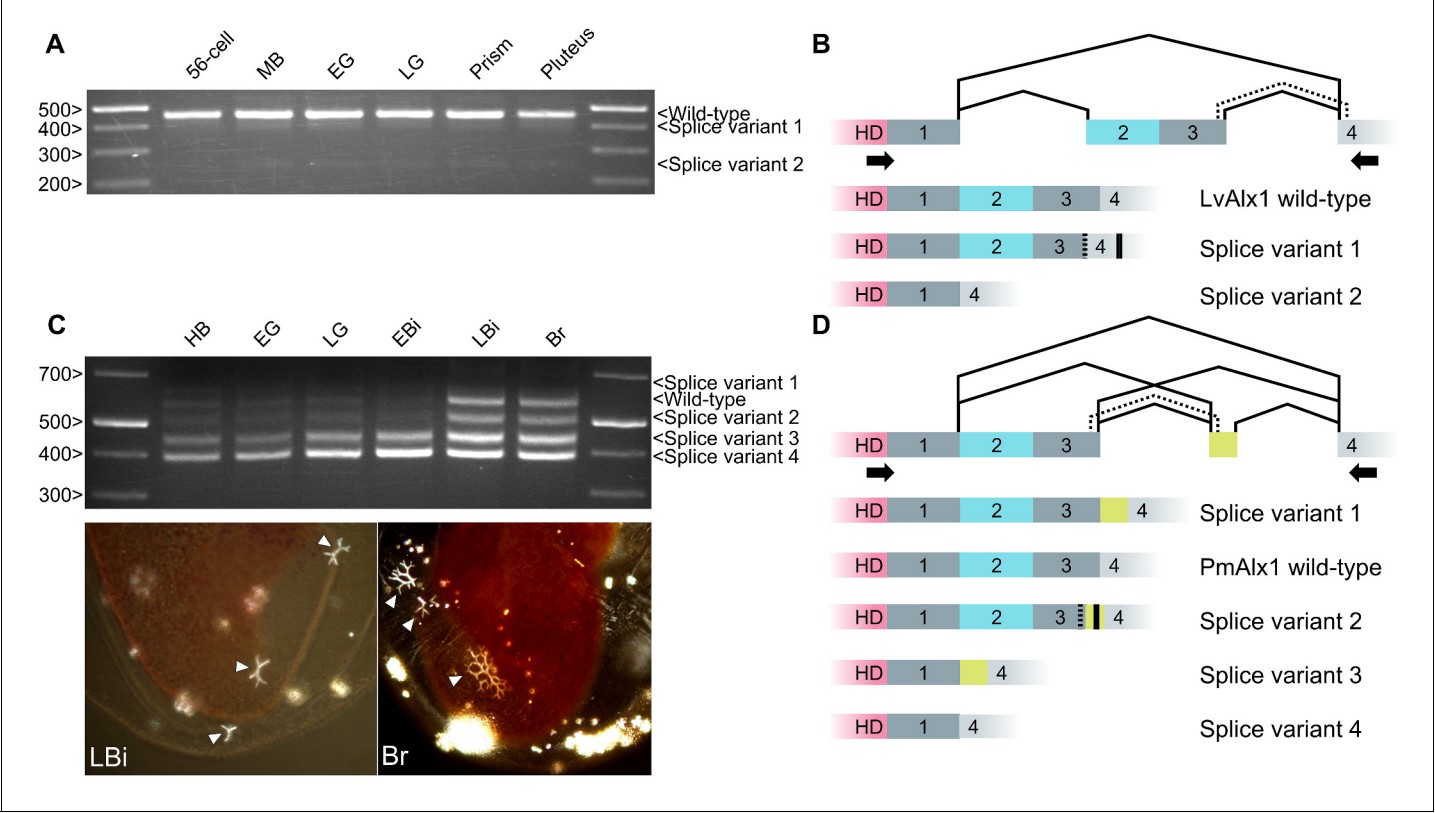

**Figure 10.** Inclusion of Domain 2 is regulated by alternative splicing. (**A**) Agarose gel showing different LvAlx1 splice forms. Wild-type LvAlx1 is the predominant splice form throughout embryonic development. MB: mesenchyme blastula; EG: early gastrula; LG: late gastrula. (**B**) Sequencing results show the inclusion of the exon containing Domains 2 and 3 in LvAlx1 splice variant 1. Cryptic splice sites (dotted line) in the Domain 3 and Domain 4-containing exons are utilized, however, resulting in a frameshift and premature stop codon (solid line). In LvAlx1 splice variant 2, the Domain (2 + 3) exon is skipped. The reading frame of Domain 4 is not altered. (**C**) Top: agarose gel showing different PmAlx1 splice forms. There are five PmAlx1 splice variants across the developmental stages tested. HB: hatched blastula; EG: early gastrula; LG: late gastrula; EBi: early bipinnaria larva; LBi: late bipinnaria larva; Br: brachiolaria larva. Bottom: images of living, late bipinnaria and brachiolaria larvae, viewed with partially crossed polarizers to highlight skeletal elements (arrowheads). (**D**) Sequencing results show the inclusion of an alternative exon (shown in purple) between Domains 3 and 4 in PmAlx1 splice variant 1. In PmAlx1 splice variant 2, the same alternative exon is included, but cryptic splice sites (dotted line) in Domain 3- and Domain 4-containing exons are utilized, causing a frameshift and resulting in a premature stop (solid line). In PmAlx1 splice variant 3, an internal splice site near the 5' end of Domain 2 is utilized and Domains 2 and 3 are skipped, while the alternative exon is included. In PmAlx1 splice variant 4, a cryptic splice site in Domain 2 is utilized and Domains 2 and 3 and the alternative exon are skipped. Arrows indicate the positions of PCR primers flanking the regions of interest.

DOI: https://doi.org/10.7554/eLife.32728.016

## Discussion

### Structure-function analysis of Alx1

Our studies show that a short form of Alx1 containing only the homeodomain and Domains 1 and 2 is able to substantially rescue PMC specification and skeletal morphogenesis in sea urchin (*L. variegatus*) embryos. Thus, the N-terminal region (which contains a consensus MAPK phosphorylation site), the conserved OAR domain, and much of the C-terminal region are largely dispensable for these functions. Studies on vertebrate Alx1 have reported that the OAR domain and N-terminal region have modest, attenuating and enhancing influences, respectively, on transcriptional activity (*Amendt et al., 1999*; *Horiguchi and Takeshita, 2003*; *Norris and Kern, 2001*). Our findings do not rule out subtle effects of these or other Alx1 domains on transcriptional activity, which might be detectable only through biochemical assays. Our analysis suggests, however, that any such effects play a relatively minor role in regulating Alx1 function in vivo. It should be noted that our rescue

assay specifically examined the embryonic function of Alx1, and the OAR domain, N-terminal region, or other domains of Alx1 may play important roles in adult skeletogenesis.

We have shown that the C-terminal region between the homeodomain and the OAR domain harbors a previously uncharacterized domain, Domain 2, that is critically important for Alx1 function in the embryo. We did not identify motifs within this short (41-amino acid) domain, nor did we find signatures of this domain in other proteins, either within or outside echinoderms. Domain 2 is highly conserved among echinoderms, however, which is consistent with its functional importance. The specific biochemical function of Domain 2 remains to be elucidated. Owning to its proximity to the homeodomain, this domain may alter DNA binding directly or may act indirectly by recruiting protein partners that influence the transcriptional activity of Alx1.

## Taxon-specific changes in Alx1

PmAlx1 and LvAlx1 share a highly conserved homeodomain and OAR domain, in addition to Domain 2. Expression of PmAlx1.WT.GFP in LvAlx1 morphants, however, did not rescue PMC specification, demonstrating that these proteins are not functionally interchangeable. Ectopic expression of PmAlx1 in *S. purpuratus* embryos has been reported to result in ectopic skeletal spicule formation and a radialized skeletal phenotype (*McCauley et al., 2012*) and we observed the same phenotype in *L. variegatus* embryos (data not shown). This result was previously interpreted as evidence of the functional conservation of the PmAlx1 and LvAlx1 proteins, but a radialized skeleton can also result from a disruption of ectodermal patterning (*Bergeron et al., 2011*; *Coffman et al., 2014*; *Coluccio et al., 2011*; *Flowers et al., 2004*). We showed that a chimeric form of PmAlx1 that contained the LvAlx1 C-terminal region (PmAlx1.LvAlx1C.GFP) was able to rescue LvAlx1 morphants, indicating that differences in the C-terminal region of PmAlx1 compromised its activity in the sea urchin embryo. These differences lie outside Domain 2, as we found that Domain 2 of PmAlx1 was functionally interchangeable with Domain 2 of LvAlx1.

## Functional divergence of Alx1 and Alx4

Functional diversification following gene duplication has been documented in several transcription factor families, such as the MADS box family in plants, the *hox3/zen/bcd* family in arthropods, and the *myb* gene family in vertebrates [reviewed by (*Lynch and Wagner, 2008*)]. Whether the functional specialization of paralogous transcription factors has played a major role in the evolution of novel structures remains an open question.

The echinoderm *alx1* and *alx4* genes, which are physically adjacent to one another, originated from a gene duplication event. *Koga et al., 2016* concluded that this duplication occurred after the divergence of echinoderms from hemichordates and was present in the last common ancestor of all modern echinoderms at least 520 million years ago. This conclusion was based on two observations: (1) all echinoderms have clear orthologues of *alx1* and *alx4* (also known as *calx*), and (2) the hemichordate *Saccoglossus kowalevskii* has a single *alx4*-like gene. The publicly available genome sequences of a different hemichordate, *Ptychodera flava* (http://marinegenomics.oist.jp/) confirms the presence of a single Alx4-like protein that clusters with the *S. kowlavskii* protein (data not shown).

The function of Alx4 in early sea urchin embryonic development has not been tested experimentally. Similarities in the amino acid sequences of Alx1 and Alx1, however, as well as their overlapping expression in PMCs and the positive regulation of *alx4* by *alx1* (*Rafiq et al., 2014*) all pointed to possible functional redundancy of the two proteins in PMC specification. Our findings clearly show, however, that Alx1 and Alx4 are not functionally interchangeable. The lack of rescue by LvAlx4 is not attributable to direct binding of the LvAlx1 MO to LvAlx4.WT.GFP mRNA, as the N-terminal MO target sequence is not conserved in *Lv-alx4*. Most strikingly, insertion of the LvAlx1 Domain 2 into LvAlx4 is sufficient to confer function and fully rescue PMC specification, patterning, and skeleton formation in LvAlx1 morphants. This observation strongly supports the view that the gain of Domain 2 was an important evolutionary innovation that marked the functional divergence of Alx1 from Alx4 by conferring skeletogenic capabilities on Alx1.

## Evolution of domain 2

Protein domains are acquired through several mechanisms, such as gene fusion, intron recombination, exon recombination, retrotransposition and exon extension (*Buljan et al., 2010*; *Marsh and Teichmann, 2010*). By comparing the intron-exon organization of *alx1* and *alx4* in echinoderms and hemichordates, we have shown that the evolution of the *alx1* paralogue was accompanied by an increase in the structural complexity of the gene. This is consistent with a general tendency for duplicated genes to diverge structurally (*Xu et al., 2012*). Several lines of evidence support the view that this structural diversification included the exonization of Domain 2 from previously non-coding sequences: (1) The lack of an identifiable Domain 2 in proteins other than Alx1 argues against the possibility that this domain originated from a donor gene, although sequence divergence could confound this analysis. (2) Domain 3 within the Domain (2 + 3) exon is intrinsically disordered, a characteristic of exons that originated from non-coding sequences (*Buljan et al., 2010*; *Marsh and Teichmann, 2010*; *Wilson et al., 2017*). (3) The complex patterns of alternative splicing of the Domain (2 + 3) exon within the echinoderms are consistent with the observation that new exons are alternatively spliced at much higher frequency than old exons (*Sorek, 2007*). (4) In asterozoan *alx1* genes, Domains 2 and 3 of Alx1 are encoded by a continuous open reading frame that also encodes a portion of the homeodomain. Based on these considerations, we propose a model whereby Domain 2 was originally acquired via recruitment of intronic sequences in an ancestral form of *alx1* (*Figure 11*). We propose that asterozoan *alx1* genes have retained the intron-exon structure of the ancestral *alx1* gene while echinozoan *alx1* genes underwent multiple rounds of intron gain following the exon extension event, including the gain of an intron that separated the Domain 2-containing exon from its parental exon (*Figures 9* and *11*). We have examined the 5' region of the intron downstream of Domain 1 in modern echinoderm *alx4* and hemichordate *alx4*-like genes and have detected no signature of Domain 2 at either nucleotide or protein level, but it seems highly unlikely that such signatures would have persisted over the >400 million years since the putative exon extension occurred. We also propose that the intrinsically disordered state of Domain 3 is a vestige of its origin from non-coding sequence (see *Wilson et al., 2017*), while the highly conserved Domain 2 has become more ordered over time in concert with its acquisition of an important developmental function.

The presence of a well-conserved Domain 2 in the Alx1 protein of sea stars, which do not form a larval skeleton, suggests that this domain may have both embryonic and adult functions. Indeed, our analysis of *alx1* splicing in *P. miniata* revealed a developmentally regulated pattern, with increased

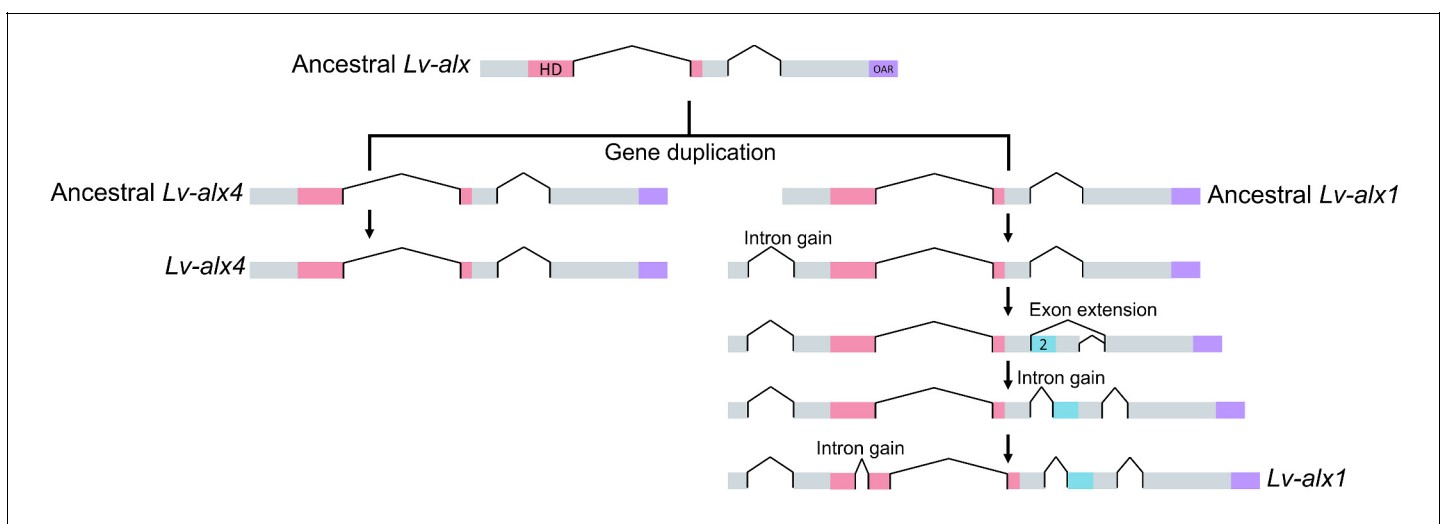

**Figure 11.** Model representing the evolution of *L. variegatus alx* genes. Following gene duplication, the ancestral *Lv-alx1* underwent rapid evolution through multiple intron gains and more importantly, acquired Domain 2 through exonization of previously non-coding sequences. By contrast, *Lv-alx4* retained an intron-exon structure very similar to that of the ancestral *Lv-alx4*.
DOI: https://doi.org/10.7554/eLife.32728.017

levels of Domain 2-containing mRNA at stages of overt skeletogenesis. Moreover, our cross-species studies demonstrated that the sea star Domain 2 is functional in sea urchin embryos. It was shown previously that in sea stars, levels of *alx1* mRNA are extremely low in the embryo but increase at larval stages, when *alx1* is expressed in the skeletogenic centers of the adult rudiment (*Koga et al., 2016*). Our findings indicate that, in addition to this control mechanism, regulation of the alternative splicing of *alx1* also influences the levels of functional (Domain 2-containing) Alx1 protein during development. Finally, we note that while a role for Domain 2 in adult skeletogenesis seems at odds with our finding that this domain is absent from the single Alx1 protein sequence from crinoids, very limited cDNA sequence data (and no genomic data) are available from this group, and additional splice isoforms may well exist.

In hemichordates, the closest outgroup to echinoderms, the single *alx4*-like gene lacks Domain 2. Adult hemichordates form small, calcareous skeletal elements termed ossicles, and orthologues of several echinoid biomineralization genes are transcribed during development (*Cameron and Bishop, 2012*). This raises the possibility that the ancestral *alx4*-like gene provided positive inputs into a 'primitive' skeletogenic GRN similar to that present in modern hemichordates. According to this view, gene duplication and the subsequent gain of Domain 2 by Alx1 altered its transcriptional activity, thereby modifying a rudimentary, ancestral skeletogenic GRN in ways that promoted the formation of much more elaborate endoskeletal system, first in the adult and later in the embryo. To explore this model further, analysis of the adult expression and transcriptional targets of the single *alx4*-like gene in hemichordates (and of the *alx4* gene in sea urchins) would be extremely informative. In addition, analysis of the biochemical function of Domain 2 will provide important insights concerning the mechanisms by which protein domain gain has driven the acquisition of novel structures.

## Materials and methods

### Animals

Adult *Lytechinus variegatus* were acquired from the Duke University Marine Laboratory (Beaufort, NC, USA) and from Pelagic Corp. (Sugarloaf Key, FL, USA). Adult *Strongylocentrotus purpuratus* and *Patiria miniata* were acquired by Patrick Leahy (California Institute of Technology, USA). Gamete release in sea urchins was induced by intracoelomic injection of 0.5 M KCl and embryos were cultured in artificial seawater (ASW) at 19–24°C (*L. variegatus*) or at 15° (*S. purpuratus*) in temperature-controlled incubators.

### DNA constructs

Total RNA was isolated from mesenchyme blastula stage embryos using the NucleoSpin RNA isolation kit (Macherey-Nagel, Germany) and cDNA was synthesized using the SuperScript IV First-Strand Synthesis System (Invitrogen/ThermoFisher, Waltham, MA, USA). Full-length *L. variegatus alx1* (*Lvalx1*) was amplified by PCR and cloned into the BamHI and ClaI restriction sites of the pCS2 +GFP vector (LvAlx1.WT.GFP). Similar steps were taken to clone *S. purpuratus* and *P. miniata alx1*. The accession numbers or IDs of previously identified *alx1* and *alx4* genes can be found in (*Ettensohn et al., 2003*; *Koga et al., 2016*; *McCauley et al., 2012*). *E. tribuloides* and *O. spiculata alx1* and *alx4* sequences were identified from databases available at http://www.echinobase.org/ (*Kudtarkar and Cameron, 2017*). *P. flava alx* sequence was identified from a database available at http://marinegenomics.oist.jp/ (*Simakov et al., 2015*). Deletion and truncation mutations were generated in *Lvalx1* using inverse PCR. To delete the desired regions, primers with overhangs containing unique restriction sites were used to amplify the entire plasmid while excluding the sequences to be removed. The PCR product was then digested with DpnI to remove the methylated parental template. The linear amplicon was digested with the corresponding restriction enzyme and self-ligated. Point mutations were generated using the QuickChange II Site-Directed Mutagenesis Kit (Agilent, Santa Clara, CA, USA). Overlap extension PCR was used to generate chimeric constructs. Individual DNA fragments were first amplified from plasmids using primers with overhangs that corresponded to the subsequent piece of DNA. Parental templates were then digested with DpnI. A second round of PCR was then carried out to assemble a full-length chimeric construct, which was subsequently cloned into the pCS2+GFP vector.

## Morpholino oligonucleotide (MO) and mRNA microinjections

Microinjection of morpholinos (MO) (Gene Tools, LLC, Philomath, OR, USA) into fertilized eggs was performed as described (*Cheers and Ettensohn, 2004*). The translation-blocking LvAlx1 MO was complementary to the 5' end of the coding region and had the sequence 5'-ACGGCATTGACGGG TAGAATAACAT-3'. This MO has been characterized in previous studies (*Ettensohn et al., 2003*; *Rafiq et al., 2014*). Plasmids were linearized and used as templates to produce 5' capped mRNA using the mMessage mMachine SP6 Transcription Kit (Ambion/ThermoFisher, Waltham, MA, USA). From 4 µg of DNA template, an average mRNA yield of 90 µg was obtained. All injection solutions contained 20% glycerol (vol/vol) and 0.1% Texas Red dextran (wt/vol) in RNAse-free, sterile water. For the LvAlx1 rescue assays, injection solutions contained of 2.0 µg/µL mRNA and 3.0 mM LvAlx1 MO. For mutant mRNAs of different lengths, the amounts of mRNAs injected were adjusted to match the molar concentration of 2.0 µg/µL of LvAlx1.WT.GFP. For observing GFP-fusion protein expression during early developmental stages, injection solutions contained 2–4 µg/µL mRNA.

## Immunostaining

PMCs were visualized by immunostaining with monoclonal antibody (mAb) 6a9, which recognizes PMC-specific cell surface proteins of the MSP130 family (*Ettensohn and McClay, 1988*; *Illies et al., 2002*). Injected embryos at the mid- or late gastrula stage were sorted by observing Texas Red dextran fluorescence under a dissecting microscope equipped for epifluorescence. The embryos were transferred to round-bottom 96-well plates for fixing and immunostaining. Embryos were fixed in 2% paraformaldehyde (PFA) in ASW for 1 hr, rinsed with ASW, and incubated in 100% methanol at −20°C for 10 min. The fixed embryos were then washed three times in phosphate-buffered saline (PBS), blocked in 5% goat serum in PBS (5% GS-PBS) for 45 min, and incubated in full-strength tissue culture supernatant containing the 6a9 antibody overnight at 4°C. The following day, the embryos were washed five times in PBS with 0.1% Tween-20 (PBST), once with PBS, and once with 5% GS-PBS. The embryos were then incubated in Alexa 488 goat anti-mouse IgG and IgM (Jackson ImmunoResearch, West Grove, PA, USA), diluted 1:50 in 5% GS-PBS at room temperature for 2–4 hr. They were washed five times with PBST and once with PBS, then were mounted on slides in anti-fade solution (DABCO) for examination.

## Light microscopy

Living embryos expressing GFP-tagged LvAlx1 (LvAlx1.GFP) were examined in their injection dishes using a compound epifluorescence microscope equipped with a 20X water-immersion objective. For observation of late stage larval skeletogenesis, living embryos were held between coverslips and microscope slides with strips of double sided tape as spacers and photographed using differential interference contrast (DIC) optics. Fixed embryos were examined using 20X and 40X dry objectives. Images were processed using ImageJ (NIH) and Adobe Photoshop CC 2015. The integrated density values of GFP fluorescence were obtained using ImageJ and subsequently normalized with the integrated density values of Texas Red dextran to account for differences in total embryo area and injection volume.

## RT- PCR

Total RNA was isolated from embryos of different developmental stages and cDNA was synthesized. Specific primer sets for *Lvalx1* and *Pmalx1* splice variants were designed to amplify products 250–650 bp in length. To sequence the splice forms, the PCR products were run on 1% agarose, extracted from the gel, and cloned into the pCS2+ vector.

## Acknowledgements

This work was supported by National Science Foundation Grant IOS-1354973 to CAE. The funders had no role in study design, data collection and analysis, decision to publish or preparation of the manuscript. The authors thank Drs. Greg Cary and Veronica Hinman for their assistance with starfish embryo culture.

## Additional information

### Funding

| Funder | Grant reference number | Author |
| --- | --- | --- |
| National Science Foundation | IOS-1354973 | Charles A Ettensohn |

The funders had no role in study design, data collection and interpretation, or the decision to submit the work for publication.

### Author contributions
Jian Ming Khor, Conceptualization, Formal analysis, Validation, Investigation, Visualization, Methodology, Writing—original draft, Writing—review and editing; Charles A Ettensohn, Conceptualization, Supervision, Funding acquisition, Methodology, Writing—original draft, Project administration, Writing—review and editing

### Author ORCIDs
Jian Ming Khor (iD) http://orcid.org/0000-0002-1428-6770
Charles A Ettensohn (iD) http://orcid.org/0000-0002-3625-0955

### Decision letter and Author response
Decision letter https://doi.org/10.7554/eLife.32728.020
Author response https://doi.org/10.7554/eLife.32728.021

## Additional files

### Supplementary files
• Transparent reporting form
DOI: https://doi.org/10.7554/eLife.32728.018

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
