## [Decision Letter]

Thank you for submitting your article "Functional divergence of paralogous transcription factors supported the evolution of biomineralization in echinoderms" for consideration by *eLife*. Your article has been reviewed by two peer reviewers, and the evaluation has been overseen by Diethard Tautz as the Reviewing and Senior Editor. The following individual involved in review of your submission has agreed to reveal their identity: Hiroshi Wada (Reviewer #2).

The reviewers have discussed the reviews with one another and the Reviewing Editor has drafted this decision to help you prepare a revised submission.

Summary:

The paper reports analysis of the transcription factor Alx1 in which the authors have experimentally analyzed a protein domain that is critical to initiating skeletogenesis in urchin embryos. By producing chimeric versions of Alx, in which the critical domain of orthologues and paralogues are substituted the authors establish the necessity and sufficiency of the domain. An analysis of Alx gene organization and splice patterns in several related species suggests that a gene duplication event was followed by a recruitment of a new exon from intronic sequences. They conclude that the gain in protein domains provides an evolutionary mechanism for novel morphology to arise.

Both reviewers find the experiments, the writing and the conclusions compelling. There are only a few points that need clarification.

Essential revisions:

1) There is a confusing aspect of the naming of the critical domain of the Alx1 protein. The authors refer to it as a C2 domain and note that their searches indicate it is not a previously characterized protein domain. There is a C2 protein domain that is well characterized and involved in calcium dependent membrane binding of proteins such as PTEN and PKC to specific membranes. As the authors do not refer to this literature, I assume the name they have chosen for the Alx1 protein domain is redundant and may confuse readers. It is therefore necessary to use another nomenclature for the domains to avoid such a confusion.

2) It is not clear how the authors established the domain boundaries. It is unclear whether they are based upon protein structure predictions or the position of the splice boundaries. Given that the minimal construct is the homeodomain+C1+C2, is it not possible that there are regions within C1 that are critical to Alx1 function? The Alx4 with the Alx1 C2 domain construct rescues, but the C1 domains of Alx1 and Alx4 are very similar (20/30 amino acids are identical). As well, the authors conclude (in subsection “Alx1 orthologues from closely related sea urchin species, but not from more distantly related echinoderms, are functionally interchangeable”) that sequence differences in the PmAlx1 outside the C2 domain are critical to its ability to rescue. It seems the C2 domain is necessary, but C1+C2 is sufficient for rescue. The concern here is that the model proposed focuses on the acquisition of the C2 domain, which appears to be defined by splice boundaries. The argument that the domain is acquired by a change in the splice location becomes somewhat circular when the domain is defined only by splice locations.

3) In the second paragraph of subsection “The C2 domain is subject to alternative splicing and likely arose by exonization”, the authors describe an analysis that suggests that C3 is intrinsically disordered, but it is not clear how they relate this to the evolution of the C2 domain. It would be necessary to show this in a specific figure and put it into the context of de novo evolution of genes (e.g. one of the latest papers on the topic is: Wilson et al., (2017).

4) Experiments are reported that indicate that loss of MAPK target sites has no effect on Alx1 function. Previous studies have established that loss of MAPK pathway components inhibits skeletogenesis. It will be necessary to have a discussion of their interpretation of these data.

---

## [Author Response]

Essential revisions:1) There is a confusing aspect of the naming of the critical domain of the Alx1 protein. The authors refer to it as a C2 domain and note that their searches indicate it is not a previously characterized protein domain. There is a C2 protein domain that is well characterized and involved in calcium dependent membrane binding of proteins such as PTEN and PKC to specific membranes. As the authors do not refer to this literature, I assume the name they have chosen for the Alx1 protein domain is redundant and may confuse readers. It is therefore necessary to use another nomenclature for the domains to avoid such a confusion.

We thank the reviewer for pointing out that “C2” was an inappropriate choice for a domain name, as that name is already associated with a well-characterized (and completely different) protein domain. Instead of calling the four regions within the C-terminal region of Alx1 “Domains C1, C2, C3, and C4,” we now assign them the very generic names of “Domains 1, 2, 3 and 4.”

2) It is not clear how the authors established the domain boundaries. It is unclear whether they are based upon protein structure predictions or the position of the splice boundaries.

The boundaries were based partly on splice junctions and partly on evolutionary conservation, not on protein structure prediction. We have clarified this in the text (subsection “Domain 2 of LvAlx1 is a novel functional domain” and the legend to Figure 1).

Given that the minimal construct is the homeodomain+C1+C2, is it not possible that there are regions within C1 that are critical to Alx1 function? The Alx4 with the Alx1 C2 domain construct rescues, but the C1 domains of Alx1 and Alx4 are very similar (20/30 amino acids are identical). As well, the authors conclude (in subsection “Alx1 orthologues from closely related sea urchin species, but not from more distantly related echinoderms, are functionally interchangeable”) that sequence differences in the PmAlx1 outside the C2 domain are critical to its ability to rescue. It seems the C2 domain is necessary, but C1+C2 is sufficient for rescue. The concern here is that the model proposed focuses on the acquisition of the C2 domain, which appears to be defined by splice boundaries. The argument that the domain is acquired by a change in the splice location becomes somewhat circular when the domain is defined only by splice locations.

The relevant experiment here is the one in which we deleted the C1 (now Domain 1) region and showed this has no effect on Alx1 function (see Figure 5). Based on this experiment, we conclude that the C1 region does not contain essential sequences. Given this finding, we do not believe there is circularity in our other arguments, as there is experimental evidence that the C2+C3 exon was gained through changes in splicing, and separate experimental data showing that C2 domain is crucial for Alx1 function.

3) In the second paragraph of subsection “The C2 domain is subject to alternative splicing and likely arose by exonization”, the authors describe an analysis that suggests that C3 is intrinsically disordered, but it is not clear how they relate this to the evolution of the C2 domain. It would be necessary to show this in a specific figure and put it into the context of de novo evolution of genes (e.g. one of the latest papers on the topic is: Wilson et al., (2017).

We thank the reviewer for calling our attention to the recent Wilson et al., (2017) paper, which we now cite. Essentially our argument is that the high degree of intrinsic disorder of the C3 domain supports the hypothesis that the C2+C3 exon arose relatively recently from non-coding sequences. This is consistent with several studies (including the Wilson paper) which show that young proteins (or protein domains) show elevated levels of intrinsic disorder, perhaps because a disordered structure makes new protein sequences less likely to be harmful to cells. We hypothesize that the highly conserved C2 domain has become less disordered over time as it has acquired an important function, while the disorder of C3 has persisted. Our feeling is that this argument is probably not complex enough to warrant a separate figure (as it is just a hypothesis), but we have expanded upon it in the Discussion section.

4) Experiments are reported that indicate that loss of MAPK target sites has no effect on Alx1 function. Previous studies have established that loss of MAPK pathway components inhibits skeletogenesis. It will be necessary to have a discussion of their interpretation of these data.

Previous studies have shown that MAPK-mediated phosphorylation of a different PMC transcription factor, Ets1, is required for PMC specification. We wondered whether MAPK might also phosphorylate Alx1 and that this might be part of the role of MAPK, but this does not seem to be the case as we can delete the phosphorylation site without affecting Alx1 function. Thus, Ets1 might completely account for the MAPK requirement. We have added statements to this effect in the text (subsection “The N-terminal region of LvAlx1 is dispensable for skeletogenic function”).